# Public-Private Partnership Transformation and Worker Satisfaction: A Case Study of Sanitation Workers in H-City, China

**Weixia Lyu** [1,†]**, Yanan Zheng** [1,†] **, Camila Fonseca** [2,†] **and Jerry Zhirong Zhao** [2,*,†,‡]

1    School of Government, University of International Business and Economics, Beijing 100029, China; lvweixia@uibe.edu.cn (W.L.); zhengyanan1994@foxmail.com (Y.Z.)

2    Humphrey School of Public Affairs, University of Minnesota, Minneapolis, MN 55455, USA; fonse024@umn.edu

\*    Correspondence: zrzhao@umn.edu

†    These authors contributed equally to this work.

‡    Current address: Humphrey School of Public Affairs, University of Minnesota, 301 19th Avenue South, Minneapolis, MN 55455, USA.

**Abstract:** Recent years have witnessed a rapid development of Public-Private Partnerships (PPP) as a new model of public service provision. Transitioning from bureaucrat- to market-oriented management of public services entails organizational changes that may affect employee satisfaction, and thus, PPP performance. We take sanitation services in H-City as a case study to explore the managerial factors that influenced worker satisfaction during the PPP transformation. Our research shows that motivation and transition factors influence worker satisfaction in the PPP transformation and may allow a smoother transformation of sanitation services. In particular, focusing on balancing workload and compensation, training, improving public attitudes, and adopting worker-friendly rules contribute to the satisfaction of sanitation workers. These findings will contribute to the transformation of the provision of public services in China.

**Keywords:** Public-Private Partnership; worker satisfaction; sanitation worker; PPP reform

## 1. Introduction

Public-Private Partnerships (PPP) have made rapid progress as a new model of service provision in recent years. China, in particular, has vigorously promoted a PPP reform, dubbed as "collaboration and societal capital," since 2014 [1]. The term "societal capital" indicates that the Chinese government collaborates not only with the private sector but also with state-owned enterprises (SOEs), which play an important role in China's market economy. Since then, the cooperation between government agencies and public- or private-enterprises has become a major engine in market-oriented reforms of public services in the country. Services that used to be provided by government agencies are now provided through PPPs. These reforms were adopted with the hope that this innovative approach will not only bring private resources to sustain China's infrastructure boom but also increase the efficiency of service provision [2]. By the end of 2017, the Chinese government had initiated more than 14,000 PPP projects [2], across a wide range of fields including economic infrastructure such as roads, bridges, and environmental facilities [3], as well as social services such as education, health care, and senior services [4].

Transitioning from bureaucrat- to market-oriented management of public services is still a key challenge for urban governance in China. Several studies reveal that labor problems are present throughout the adoption and implementation of PPPs [5–7]. These problems have been particularly

concerning in sanitation services, even before the PPP reform. Sanitation workers have had low wages, insufficient welfare coverage, high workloads, and have faced higher job risks and higher discrimination from other citizens [8]. Addressing these factors may lead to a smoother PPP transformation.

The purpose of this study is to explore the managerial factors that affected worker satisfaction in the PPP transformation of sanitation services. We build on the traditional literature on job satisfaction that emphasizes motivation factors. We test for the effect of these factors and propose additional transition factors that affect employee satisfaction in the transformation of sanitation services. Factors that affect job satisfaction impact the PPP performance and may lead to a smoother PPP transformation. We take H-City, a major city in southern China, as a case study for this research. H-City's transformation was comparatively smooth relative to other cities in China, for instance, H-City had no sanitation worker strikes. Additionally, the PPP transformation in H-City is well suited to study job satisfaction among sanitation workers, as the PPP maintained the employment of sanitation workers previously employed by the city.

H-City started a PPP transformation to provide citywide sanitation services in late 2015, striving to build itself into a "National Civilized City" and "National Health City" (Shuangchuang). This PPP transformation entailed the creation of a special purpose vehicle (SPV), a joint venture that serves as the platform of partnerships under the supervision of the Department of Sanitation of the city. In the SPV, the municipal government holds 30 percent of the equity, while the remaining 70 percent is held by three private contractors, including a private company and two SOE companies. The contractors took over sanitation services of roads, green spaces, water areas, public restrooms, and waste collection and transfer; while the city's government focused on the inspection, assessment, and supervision of the contractors' daily operations. H-City's government established a scoring system to assess the quality of sanitation. If a company scores less than 90 points in the quarterly assessment, the contracted payment is reduced. The city was successfully awarded the title of "Shuangchuang" in 2017.

The rest of the paper is organized as follows. We first present a review of the literature. In Section 3, we present the theoretical framework of hygiene, motivation, and transition factors that affect worker satisfaction. We discuss the data and methodology in Sections 4 and 5, respectively. In Section 6, we present the factors influencing sanitation worker satisfaction. Our results suggest that motivation and transition factors influence worker satisfaction in the PPP transformation. Finally, we conclude in Section 7 with a summary of our findings, their theoretical, practical and policy implications, as well as possible directions for future research.

## 2. Literature Review

Public-Private Partnerships (PPPs) are long-term contractual collaborations between the government and the private sector in the areas of infrastructure and provision of public services. Proponents of this mechanism argue that PPP offers Value for Money (VfM)—an advantageous combination of benefits and costs—in public service delivery, as governments access capital to fund investments, transfer risks, and bring technology and efficiency into the provision of public services [9,10]. Many counties, particularly China, have used PPPs in a wide range of areas including transportation, public education, public health, and more recently in water and sanitation. Research in most of these areas is extensive [11,12] but not for sanitation, which is typically limited to solid waste management [13–15]. We contribute by focusing on understudied sanitation PPPs, which entails the cleaning of roads and public restrooms in addition to managing solid waste collection and disposal.

While PPPs have long been a topic of interest in public management, the literature has largely focused on its adoption and paid much less attention to its management and performance. In the United States, research shows that the adoption of PPP is motivated by traffic demands and fiscal pressures, but resisted by liberal political ideology and administrative inertia [16]. In the Chinese context, scholars find that only a small percentage of PPP projects have reached deals. The chance for a project to successfully sign a contract within one year of its initiation is significantly affected by

transaction costs related to the government, the market, the operating environment, and project-level characteristics [1]. The duration that is required for a PPP project to reach a deal is affected by fiscal capacity and managerial capacity of local government, as well as leadership characteristics and project features [17].

With respect to the limited literature on PPP performance, existing studies are mostly about external performance, that is, the extent to which the PPP reaches intended goals or outcomes. Some research evaluates PPP performance based on their capacity to solve financial difficulties, effectively allocate resources, and improve accessibility, affordability, governance, and customer satisfaction [18–20]. For example, Wang and Zhao [10] apply a goal-centered approach to evaluate the effectiveness of nine PPP highway projects in Virginia, USA, that were implemented and opened to traffic between 1990 and 2016. These PPP arrangements have been effective in accessing innovative finance and preventing cost overrun, while the evidence is limited regarding shifting revenue risk or achieving efficiency gains. Others evaluate PPP in terms of cost-saving, externalities, fairness, accessibility, participation, and democracy [5].

Only in recent years, researchers have taken a closer look at internal processes of PPPs and their impacts on PPP performance [21]. Several studies on this area focus on PPP partners and highlight the importance of developing local human resources to take management responsibilities, creating communication strategies, and having clear roles and responsibilities across partners along with their willingness to compromise [22–24] in order to have successful PPPs. Warsen et al. [25] find that a high level of trust between partners is very important to ensure a smooth process and good cooperation in PPPs. Research regarding employees in the context of PPP reforms is limited, in particular about employee satisfaction, even though it is one of the key factors that influence the performance of an organization. High levels of job satisfaction are related to positive work outcomes and lower staff turnover [26,27], and contribute to the smooth introduction of organizational changes [28].

We will focus on particular parts of the internal process that affect worker satisfaction in the context of the transformation of a labor-intensive service. These are important because PPP reforms entail organizational changes that may affect employee satisfaction and result in worker strikes, which once happened in the PPP transformation of sanitation services. Guangzhou City reported several collective strikes of sanitation workers in the early 2010s. Workers complained about PPP transformation that failed to honor or timely fulfill the contractual terms related to wages, benefits, and workload [29]. In 2013, the city took back some sanitation services that were previously outsourced to a private cleaning company [30,31].

## 3. Theoretical Framework: Factors Influencing Workers' Satisfaction in the PPP Transformation

We build on the traditional literature to define the managerial factors that influence worker satisfaction in the transformation of sanitation services. We hypothesize that the PPP transformations affect employee satisfaction through two channels: indirectly by modifying motivation factors and directly by experiencing the PPP transformation. In addition, changes in hygiene factors are not expected to affect employee satisfaction. Herzberg et al. [32] proposed the two-factor theory that argues that hygiene factors such as wages, salaries, and working environments are related to job dissatisfaction, while motivation factors such as job reputation, promotion, and recognition are associated with job satisfaction. Improving the factors that cause dissatisfaction does not create satisfaction. Therefore, changes in motivation factors due to the PPP transformation may have affected employee satisfaction and may have allowed a smoother transformation of sanitation services in H-City.

In addition to these factors, there are transition factors that may also affect the attitudes of employees towards their jobs. Employees that were hired before the PPP transformation will have different expectations of their work and may respond differently to market-based management than new employees (those hired after the PPP transformation). In particular, the experiences and expectations of employees that were involved in the PPP transformation will significantly influence

their emotions, loyalty, and job satisfaction. The transition to market-based management may lead to a change in corporate culture, higher occupational stress, and lower job satisfaction [33].

*3.1. Hygiene Factors and Sanitation Workers' Satisfaction*

**Working compensation:** The literature shows that there are significant changes in working conditions and safety measures after outsourcing public services. In particular, studies found that privatization and outsourcing result in low wages and fewer benefits for employees [34–37]. Similar results have been identified in China after the implementation of the PPP model in sanitation services [38]. Accordingly, we proposed the following hypothesis:

**Hypothesis 1 (H1).** *Annual wages do not have a significant effect on sanitation workers' satisfaction in the PPP transformation.*

**Working time arrangements:** Working time arrangements are key elements of working conditions and determine the possibilities for employees to balance work with their other life spheres [39]. In China, scholars have generally found that working time increases after a PPP transformation [40,41]. These long working hours have been found to reduce job satisfaction [42,43]. However, following the two-factor theory, we do not expect that working time arrangements, like other hygiene factors, have a significant effect.

**Hypothesis 2 (H2).** *Daily working hours do not have a significant effect on sanitation workers' satisfaction in the PPP transformation.*

*3.2. Motivation Factors and Sanitation Workers' Satisfaction*

**Perceived compensation fairness:** Employee satisfaction is affected by the perception of being paid fairly [44,45]. In this context, we refer to this as the perceptions of being compensated commensurately to their contribution to the firm's output. Therefore, we propose the third hypothesis:

**Hypothesis 3 (H3).** *Perceived compensation fairness has a positive effect on sanitation workers' satisfaction in the PPP transformation.*

**Perceived public attitude:** The literature on sanitation workers in China generally finds that urban citizens are not respectful or friendly to sanitation workers [46]. Some people, influenced by traditional views, treat sanitation workers as "secondary social groups" [47]. This lack of acceptance by the public gives sanitation workers a distinct feeling of being abandoned by society. Ye et al. [48] argue that maintaining the dignity of sanitation workers, increasing public awareness about the importance of sanitation services, and respecting the work of sanitation workers would increase their job satisfaction. Therefore, we propose the fourth hypothesis:

**Hypothesis 4 (H4).** *Perceived public attitude has a positive effect on sanitation workers' satisfaction in the PPP transformation.*

**Training:** Researchers have found that sanitation workers in China receive insufficient training [47]. However, training has been found to be positively related to job satisfaction [49,50]. When employees are trained and engaged in organizational decision-making and policy planning, executing policies become easier [51]. In PPPs, training is considered an important performance indicator that affects the process of the PPP. Employees should receive training to gain new knowledge and improve individual and organizational performance [52,53]. Therefore, we propose the fifth hypothesis:

**Hypothesis 5 (H5).** *Training has a positive effect on sanitation workers' satisfaction in the PPP transformation.*

**Perception of sanitation quality:** In sanitation services, research has found that workers' job satisfaction is significantly affected by the overall performance of sanitation services perceived by the public [54]. In this context, we take the perception of sanitation quality as the perception of the working environment. Scholars have found that having a good working environment positively affects the levels of job satisfaction [55,56]. In the context of PPPs, Wetterberg [57] finds that PPPs have the potential to enable stronger labor protections and improve working conditions. Therefore, we proposed the sixth hypothesis:

**Hypothesis 6 (H6).** *Overall perception of sanitation quality in H-City has a positive effect on sanitation workers' satisfaction in the PPP transformation.*

*3.3. Transition Factors and Sanitation Workers' Satisfaction*

**Perception of the transition process:** Studies have found that organizational changes affect job satisfaction and employee performance [58,59]. In the particular context of PPPs, Wang et al. [60] found that employees' positive perceptions of the transition process increase their commitment to the management company in new forms of the employment relationship. Therefore, we propose the seventh hypothesis:

**Hypothesis 7 (H7).** *Employees' perceptions of the transition process have a positive effect on sanitation workers' satisfaction in the PPP transformation.*

**Transition:** In the context of sanitation PPP transformation in the Dalian City, Li [61] found that after the PPP transformation there is a lack of managerial capacity—indicated by ineffective evaluation systems, and the lack of pay incentives, career development, and promotion mechanisms—that affects the satisfaction of sanitation workers. A change in managerial capacity before and after the PPP may be experienced only by sanitation workers who were employed before the transformation.

**Hypothesis 8 (H8).** *Experiencing the PPP transition has a negative effect on sanitation workers' satisfaction in the PPP transformation.*

**4. Data**

The data collection involved the administration of a survey in H-City. We developed a survey that was randomly administered to sanitation workers in H-City in 2018. Research assistants conducted individual surveys from February to August 2018 and data entry clerks generated the dataset from questionnaires using a double-check method. Participants were asked about demographics, job satisfaction, and hygiene, motivation, and transition factors. To inform the survey questionnaire, we first conducted semi-structured interviews with eight H-City government staff, fifteen sanitation workers, three sanitation companies, and the H-City sanitation PPP Platform Company.

In total, we surveyed 556 sanitation workers in H-City. All the interviewees were front-line employees, including those who clean streets and public restrooms, and those who collect and transport solid waste. Of them, 352 sanitation workers were employed before the PPP reform and 204 sanitation workers were hired after the reform. The sample rate is 5.17 percent, as there were approximately ten thousand sanitation workers in H-City [62]. In terms of hygiene factors, we asked about annual wages in RMB and daily working hours. In terms of motivation factors, we included questions related to perceived compensation fairness, perceived public attitude towards sanitation workers, training frequency, and perceptions of the overall sanitation quality in H-City. In terms of transition factors, we included questions regarding the perception of the overall effectiveness of the sanitation PPP transformation, and whether the sanitation worker was employed before the PPP transformation. Demographic questions such as age and gender were also included.

### 5. Methodology

We use an ordered choice model to analyze the factors that influenced sanitation worker satisfaction during the PPP transition in H-City. We define our dependent variable, SATISFACTION, on a scale of one to five. We asked participants to score how satisfied they were with their current job, one being "very unsatisfied" and five "very satisfied". This implies that the respondents expressed a preference with a sequential and meaningful order (ordinal ranking as in Greene [63]). Given the nature of our dependent variable, the ordered logistic regression is a suitable tool for analysis. The ordered logistic model is based on the following specification:

$$y_i^* = \beta' X_i + \varepsilon_i,$$

where $X_i$ is a set of explanatory variables and $\varepsilon_i$ is the error term. The variable $y_i^*$ corresponds to the worker's actual satisfaction, which is unobservable, instead, we observe $y_i$ that corresponds to the categories of response given various threshold points (censoring):

$$y_{i=} \begin{cases} 0 & if \ y^* \leq 0, \\ 1 & if \ 0 < y^* \leq \mu_1, \\ 2 & if \ \mu_1 < y^* \leq \mu_2, \\ \dots \\ J & if \ \mu_{J-1} \leq y^* \end{cases}$$

For our model, the explanatory variables are hygiene, motivation, and transition factors. In terms of hygiene factors, we include annual wages (WAGE) and daily working hours (WORKHOUR). In terms of motivation factors, we included the perceived compensation fairness (COMPENSATION), the perceived public attitude towards sanitation workers (ATTITUDE), training frequency (TRAINING), and perceptions of the overall sanitation quality in H-City (QUALITY). In terms of transition factors, we included the perception of the overall effectiveness of the sanitation PPP transformation (PPP EVAL) and whether the sanitation worker was employed before the PPP transformation (TRANSITION). In addition, we include demographics as previous studies have found that they affect the satisfaction of sanitation workers in the PPP reform. For instance, Ding [46] finds that age affects satisfaction through the occupational identity of sanitation workers. Table 1 presents the definitions and descriptive statistics of the variables captured in the survey. The $\mu_i$ and $\beta$ are unknown parameters to be estimated by maximum likelihood. The significance of the elements of $\beta$ will inform us about the validity of our hypothesis.

**Table 1.** Variable definitions and descriptive statistics.

| Variable | Obs | Mean | Std. Dev. | Min | Max |
|---|---|---|---|---|---|
| SATISFACTION | 556 | 3.178 | 0.888 | 1 | 5 |
| WAGE | 556 | 1.736 | 0.567 | 1 | 4 |
| WORKHOUR | 556 | 1.324 | 0.540 | 1 | 4 |
| COMPENSATION | 555 | 0.252 | 0.435 | 0 | 1 |
| ATTITUDE | 556 | 3.496 | 0.841 | 1 | 5 |
| TRAINING | 556 | 1.741 | 0.758 | 1 | 3 |
| QUALITY | 556 | 4.061 | 0.720 | 1 | 5 |
| PPP EVAL | 556 | 3.655 | 0.818 | 2 | 5 |
| TRANSITION | 556 | 0.633 | 0.482 | 0 | 1 |
| AGE | 556 | 2.629 | 0.652 | 1 | 4 |
| GENDER | 556 | 0.849 | 0.358 | 0 | 1 |
| SATISFACTION | Sanitation workers' job satisfaction: Very dissatisfied (1); dissatisfied (2); neutral (3); satisfied (4); and very satisfied (5). | | | | |
| WAGE | Annual wage in RMB: Below 1,500 (1); between 1,500 and 2,500 (2); between 2,500 and 3,500 (3); and more than 3,500 (4). | | | | |
| WORKHOUR | Sanitation workers' daily working hours: Below 8 hours (1); between 8 and 10 hours (2); between 10 and 12 hours (3); and more than 12 hours (4). | | | | |
| COMPENSATION | Employee perceives that the compensation corresponds to the workload (1); Employee perceives that the compensation does not correspond to the workload (0). | | | | |
| ATTITUDE | Public attitude towards sanitation workers: Very unfriendly (1); unfriendly (2); neutral (3); friendly (4); and very friendly (5). | | | | |
| TRAINING | Training for sanitation workers in a year: None (1); between 1 and 5 times (2); and more than 6 times (3). | | | | |
| QUALITY | Sanitation workers' perception of overall sanitation quality in H-City: Very bad (1); bad (2); moderate (3); good (4); and very good (5). | | | | |
| PPP EVAL | Sanitation workers' perception of the overall effectiveness of the PPP transformation: Very bad (1); Bad (2); Neutral (3); Good (4); and Very Good (5). | | | | |
| TRANSITION | Sanitation worker employed before the PPP transformation (1); Sanitation worker employed after the PPP transformation (0). | | | | |
| AGE | Sanitation workers' age: 20–29 years old (1); 30–39 years old (2); 40–49 years old (3); and more than 50 years old (4). | | | | |
| GENDER | Female (1); Male (0). | | | | |

## 6. Findings

Table 2 present the estimates from three models differing in sample selection. Model I includes the full sample, Model II only includes sanitation workers who have been employed before the PPP transformation, and Model III includes only sanitation workers hired after the PPP transformation. For the purpose of this research, the first model is the most important as it provides a way to compare job satisfaction between the sanitation workers hired before and after the PPP transformation and test the hypothesis related to transitional factors. The other two models are important for understanding job satisfaction within each type of employee. The chi-squares are highly significant, suggesting a good model fit across all three models.

**Table 2.** Ordered logit results.

| Variables | Model I | Model II | Model III |
|---|---|---|---|
| WAGE.2 | 0.0113 | 0.0310 | 0.0317 |
| WAGE.3 | −0.626 | −0.522 | −0.611 |
| WAGE.4 | 1.248 | 2.689 ** | −3.236 * |
| WORKHOUR.2 | −0.359 * | −0.243 | −0.687 ** |
| WORKHOUR.3 | −0.460 | −0.768 | |
| WORKHOUR.4 | 1.156 | 0.934 | 1.715 |
| COMPENSATION | 1.285 *** | 0.707 *** | 2.379 *** |
| ATTITUDE.2 | 1.483 ** | 1.641 ** | −0.127 |
| ATTITUDE.3 | 2.366 *** | 2.716 *** | 0.554 |
| ATTITUDE.4 | 2.838 *** | 3.189 *** | 1.039 |
| ATTITUDE.5 | 3.761 *** | 4.542 *** | 0.878 |
| TRAINING.2 | 0.304 * | 0.290 | 0.603 * |
| TRAINING.3 | 0.486 ** | 0.414 | 0.763 |
| QUALITY.2 | 1.797 | 1.523 | |
| QUALITY.3 | 1.703 | 1.618 | −0.345 |
| QUALITY.4 | 2.004 * | 1.769 | 0.163 |
| QUALITY.5 | 2.274 * | 2.061 * | 0.346 |
| PPP EVAL | 0.437 *** | 0.452 *** | 0.402 ** |
| TRANSITION | −0.354 ** | | |
| AGE.2 | −0.0636 | 0.472 | −0.735 |
| AGE.3 | −0.136 | 0.335 | −0.634 |
| AGE.4 | 0.0733 | 0.344 | −0.0409 |
| GENDER | −0.0189 | −0.0470 | −0.123 |
| Constant cut1 | 2.544 | 3.424 * | −1.920 |
| Constant cut2 | 4.713 *** | 5.722 *** | 0.111 |
| Constant cut3 | 6.821 *** | 7.611 *** | 2.871 |
| Constant cut4 | 9.997 *** | 11.22 *** | 5.870 ** |
| Chi2 (df) | 133.20 (23) | 89.56 (22) | 73.96 (20) |
| Observations | 555 | 351 | 204 |

Note: The first category is the reference level for all categorical variables. We present the name of the variable and a number that indicated the level of the categorical variable. Differences between the number of observations used in the regression and the reported in Table 1 differ due to missing information in the variable COMPENSATION. *** $p < 0.01$, ** $p < 0.05$, * $p < 0.1$.

Model I and Model II satisfied the proportional odds assumption. One of the underlying assumptions of the ordered logistic regression is the proportional odds assumption or the parallel regression assumption. This assumes that the coefficients that describe the relationship between each pair of outcome groups is the same for all outcome groups. We performed the Brant test and a likelihood ratio test to test this assumption. The results from both tests indicate that Model I and Model II have not violated the proportional odds assumption. This assumption is not satisfied in Model III. Therefore, we discuss our findings based on the results of the first two models.

To test H1 and H2 we look at the significance of annual wages (WAGE) and daily working hours (WORKHOUR). Neither is significant, which supports H1 and H2. As it is expected by the two-factor theory, changes in hygiene factors do not have a significant effect on job satisfaction.

In terms of motivation factors, perceived compensation fairness (COMPENSATION) and perceptions of public attitude towards sanitation workers (ATTITUDE) have a positive and significant impact on the satisfaction of sanitation workers. The results are consistent across models I and II and the findings support H3 and H4, respectively. According to our interviews in H-City, three-quarters of the sanitation workers believed that public attitudes toward their occupation had improved in the last three years, and that most residents are now more friendly and respectful. Part of the improvement in the public perception of sanitation workers can be due to the increase in public education toward sanitation services through media, pushed by the government as part of the effort to build "Shuangchuang". While in the past many residents did not support the sanitation work and littered garbage around,

after the PPP reform the city's general cleanliness and civic culture have improved, and the sanitation workers are more respected.

Training for sanitation workers (TRAINING) also has a positive and significant impact on sanitation worker satisfaction, in line with H5. This variable is only significant when considering the full sample. From the interviews, we learned that before the PPP there was almost no training. In contrast, after the PPP, training sessions were organized for workers in different jobs. For instance, jobs that require machine operations received more training, particularly, on safety.

Figure 1 shows the probability of each level of job satisfaction with respect to each level of public attitude (first column) and frequency of training (second column) at different sample sizes (in rows).

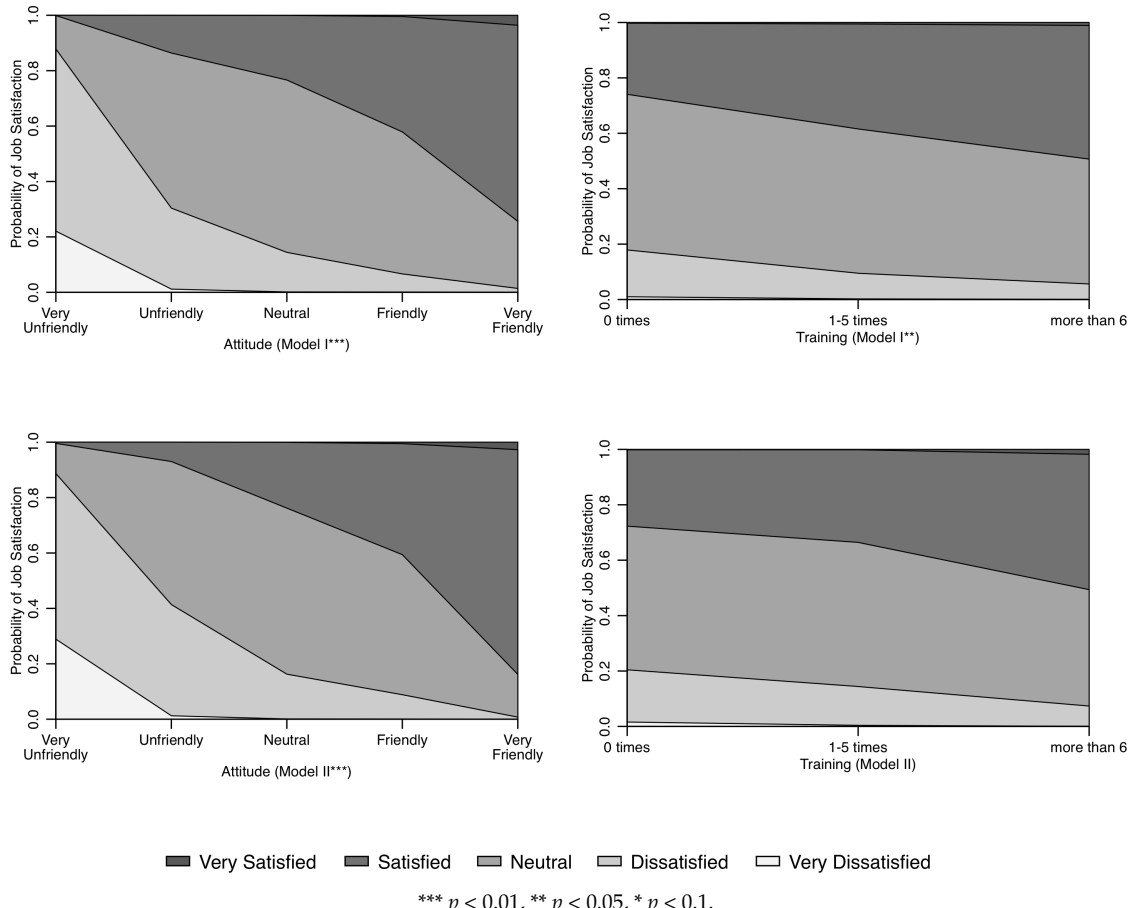

*** $p < 0.01$, ** $p < 0.05$, * $p < 0.1$.

**Figure 1.** Attitude, training and the probability of job satisfaction.

Lastly, in terms of motivation factors, the perception of sanitation workers of the overall sanitation quality in H-City (QUALITY) shows a low and borderline significant impact on sanitation worker satisfaction in Model I. Thus, H6 is not supported.

In terms of transition factors, the way sanitation workers perceived the transition process (PPP EVAL) has a significant effect on their job satisfaction, which supports H7. In addition, workers who were already employed before the transition have significantly less satisfaction than workers that were hired after it (TRANSITION), as we expected from H8. We learned from the interviews in H-City that the operation management became stricter after the PPP transformation. All three private companies imposed penalties and fines for sanitation workers that underperformed. Managers randomly checked streets and reported to the company if they found cigarette butts or other garbage. Sanitation workers who were responsible for the area were fined. According to managers, the purpose of the fine was to serve as an urge to enhance service quality and not as a punishment. In addition, sanitation workers enjoyed a more flexible working schedule in which they were able

to take breaks before the PPP transformation. The working hours became strictly supervised after the PPP transformation. One of the companies, for instance, used a GPS (Global Positioning System) bracelet to prevent sanitation workers from leaving their jobs during work hours. These tight control measures may have impacted worker satisfaction negatively. With regards to control variables, AGE and GENDER do not have a significant impact on sanitation worker satisfaction.

The results suggest that perceived compensation fairness, public's attitude toward sanitation workers and the perceptions about the transition process contribute to job satisfaction of the whole body of sanitation workers in the PPP transformation. Results from models II and III indicate that, in terms of job satisfaction, improving public's attitude towards sanitation services matters for sanitation employees who experienced the PPP transformation, while providing adequate training is important for newly hired employees.

## 7. Conclusions and Discussions

This study aimed to explore the managerial factors that affect job satisfaction in the PPP transformation. Our research shows that motivation and transition factors influenced sanitation worker satisfaction in H-City's PPP transformation. Focusing on these factors may contribute to a smooth and successful PPP transformation of sanitation services in China. Our findings contribute to the literature and have practical implications for future efforts to transform public service delivery.

First, the study contributes to the literature on PPPs by focusing on their internal management, particularly factors affecting worker satisfaction. This complements previous studies that generally focus on the external performance of PPPs. In addition, we add to the literature on the provision of public services by focusing on sanitation services. Sanitation services are understudied relative to other public services and their treatment is typically limited to waste management, even though sanitation services like road cleaning and waste collection accounted for 64 and 22 percent of the total municipal sanitation market in China in 2017, respectively [64].

Second, these findings provide useful information for PPP implementation. In terms of motivation factors perceived compensation fairness, public attitudes, and training have a significant impact on the satisfaction of sanitation workers in the PPP transition. Therefore, focusing on balancing workload and compensation, improving the public's perception of sanitation jobs, and providing training sessions can contribute to improving sanitation worker satisfaction. In terms of transition factors, there is a significant difference in satisfaction between the sanitation workers hired before and after the PPP transformation. The results suggest that companies should carry out more friendly management systems and give more care to transitioning workers to improve their job satisfaction.

This research has some limitations. First, there is limited publicly available information regarding the sanitation service sector in H-City as well as information regarding internal structural changes that came with the PPP transformation. Second, we relied on self-reported data from surveys, and thus, the study is subject to potential bias in the way workers report the information. Third, the Chinese context can make some of the conclusions hard to generalize. Even though many of the characteristics of PPPs are shared across countries, especially developing countries, some unique features of labor conditions in China, including the prominence of state-owned (and state-run) enterprises and the prohibition to layoff previous employees, gives PPPs less leeway than they would otherwise have.

Our findings in the context of sanitation services transformation may apply to other PPP transformations. In China, PPP transformations are taking place across many public service sectors, such as environmental protection, hospitals, and school logistical services [1]. This would be a fruitful area of future research. Future research may also investigate job satisfaction before and after the PPP transformation, which may allow for identifying changes in job satisfaction and their possible causes. This research could also be conducted in other countries, particularly in developing countries. Whether the findings can be further generalized to other countries remains a question until there is more evidence from comparative studies.

**Author Contributions:** Conceptualization, J.Z.Z. and W.L.; methodology, J.Z.Z. and C.F.; formal analysis, C.F. and J.Z.Z.; investigation, W.L. and Y.Z.; resources, W.L.; data curation, J.Z.Z.; writing—original draft preparation, Y.Z.; writing—review and editing, C.F. and J.Z.Z.; supervision, W.L. and J.Z.Z.; project administration, W.L.; funding acquisition, W.L. All authors have read and agreed to the published version of the manuscript.

**Funding:** This research was funded by the Beijing Social Science Fund Key Project grant number 16JDGLA021.

**Conflicts of Interest:** The authors declare no conflict of interest.

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
