# Peer review of "Public-Private Partnership Transformation and Worker Satisfaction: A Case Study of Sanitation Workers in H-City, China"

_sustainability, doi:10.3390/su12135479_

Round 1
Reviewer 1 Report
The Authors presented a very interesting but important aspect - PPP transformation and worker satisfaction. The authors conducted a case study of sanitation workers in H-city, China. I think the authors made great effort in conducting the interviews and surveys. The results are interpreted. However, there are certain limitations and issues found within this paper. During the reviewing of this paper, I have the following editorial and technical comments for the authors to consider and improve:
- Overall, the paper is well written in English. However, I still notice wording and grammar issues now and then. Therefore, I recommend the authors perform professional English proof reading before next submission
- A major limitation of the paper is that many of the discussions and facts only apply to China, which makes the paper not universal interested. For example, the working hour of a sanitation worker can exceed 12 hours per day - this may sound crazy in many countries.
- The Introduction Section is not well organized. The authors only provide the fact that PPP has made rapid progress in China. But did not present the reason why this is the case. In other words, what are the advantages of PPP scheme in China to make PPP so popular? In addition, the background information on worker satisfaction should be provided in the introduction part. Introduction Section needs to be re-organized. Please consider to revise.
- In addition, in the Introduction Section, I would expect more basic introduction of current practices of PPP in China using more numbers and facts instead of descriptive languages. There should be more discussion on general PPP practices and sanitation practices in China before talking about H-city in Line 26. Please consider to add.
- Page 2 Line 39 to Line 41. Add "," after Section X. For example, "In Section 3, we ..."
- Literature review section also needs major revision. Literature review summary is too brief and the authors should expand the literature review findings of each reference instead of putting them into categories. This will work well if you have a long literature review section. However, this manuscript only have two paragraphs of literature review - one for external performance and the other for internal process. I highly recommend that the authors provide much more details regarding each reference cited from Line 45 to Line 58 so that the readers can understand more. For example, what existing research has found regarding human resource management, etc.
- Comment #6 applies to all the references used in this paper because many references are in Chinese and it is very difficult for non-Chinese speaking readers to read.
- Page 2 Line 46. Delete "been" in the phrase "have been taken"
- Page 2 Line 59 to Line 65. Instead of literature review, I think this paragraph fits better in introduction section.
- The authors should provide the definition of "hygiene factors" and "motivation factors" somewhere in the introduction section or Section 3.
- I am not sure if the authors know that "hygiene factors" mean. According to the definition, hygiene factors are the factors that "do not give positive satisfaction or lead to higher motivation". Well, in H1, the authors hypothesized that annual wages have a positive effect. If annual wages have positive effect, then this factor can not be counted as a hygiene factor under 3.1. Why did this happen?
- Page 3. Line 105 to Line 111. The authors said "most sanitation workers believe that their long working hours are not fully compensated'. But why the hypothesis is that "Perceived compensation fairness has a positive effect"? If the workers believe their working hours are not fully compensated, how can it be a positive effect?
- Section 3 needs to re-organized as well. Section 3 mentions many times of the "interview". The authors should provide all the details about the interview at the beginning of Section 3 (instead of Section 4) to make it more logical order. Please consider to revise
- Again, those hypotheses were made based on the interviews. However, based on the information provided in Section 4, the authors interviewed only 15 sanitation workers (regardless of staff and personnel from the companies and government). Considering there are about 10,000 sanitation workers in H-city, the sample size is too small to make any hypotheses. Moreover, the authors did not provide how these 15 workers were selected and what kind of questions were asked. Therefore, these findings and hypotheses made based on such small sample seem pretty subjective and are not persuasive.
- Page 4 Line 134. "different jobs"
- Page 4 Line 168. The authors said that 15 questions were developed. However, Table 1 only contains 12 variables. Why? If the questions does not provide variables to the model, why did you ask?
- Page 5. Table 1. "Female" should be changed to "Gender"
- Page 5. Table 1. Why the number of observations in "Resident" is 546? not 556?
- Page 7. Table 2. Why the number of observations of Model I, Model II, and Model III do not match 556, 352, and 204 as provide by the authors in Line 172 to Line 173?
- Page 7 Line 207 to Line 221. These discussions corresponds to Model I. The authors should make this clear.
- The authors should pay attention to the description of the results. For example, Line 213. "Training for sanitation workers also has a positive and significant impact..." I found that "Training 3" is not significant based on Table 2. Therefore, the author's description is not accurate and should be revised.
- The authors need to pay attention to draw any conclusions based on this case study. This paper can only provide a methodological framework for similar analyses in China. I believe the findings from this paper only apply to H-city. I believe each city has its own characteristics. This should be pointed out.
- What are the limitations of this paper and research? Any future directions?
Hope my comments can help you further improve the quality of the manuscript.
Reviewer 2 Report
Comments and Suggestions for Authors
Introduction
It is recommended to state the aim of paper and expected contribution in the introduction part.
Theories and Hypothesis
On which basis, you have selected the three factors (Hygiene, motivation, and transition)? It is better to add the reason.
It is unusual to develop hypothesis on the basis of collected data (interview). It is developed on the basis of theory or literature normally. Why have you selected this approach? Please write clearly.
It is better to present a theoretical framework to show the relation between past research and current research in terms of presenting a research gap.
Data
The data is collected in two years 2007 and 2008. I suggest you writing the reason and logic behind the strategy clearly. Why is it necessary to collect data in two years?
In table 1. It is suggested to add a connection between variables and the factors.
Methodology
The validation of logit model and its specifications is missing. How to validate the mentioned equation on page 6?
Findings
I suggest you adding the description of 3 models used in this research with significance.
On page 8, figure 1. Shows the probability of each level of job satisfaction with respect to each level of public attitude and frequency of training at different sample sizes. As a suggestion, it is better to add the significance of the figure with description.
Conclusion and Discussion
Conclusions doesn't make a real academic contribution. It is recommended to add precisely.
It is suggested to describe the generalizability of the current findings to other sectors not the cities as they are significant.
It is necessary to add the possible future work in the specified research field in the end.
Reviewer 3 Report
good paper. Just a few remarks.
- refine references; you often swap first name with family name eg reg. 21 or 33
- add more recent literature, for instance digiting "ppp sanitation" in google scholar, researchgate, ssrn, scopus, isi/web of science, etc.
- is H-City a fancy denomination? clarify
- colours in black and white figure should be more distinguishable
- improve the background with a better framework of PPP (which is its purpose?). Introduce the concept of Value for Money that links private to public interests
- conclusion: why is the paper interesting? which can be new research avenues?
- state clearly which are the advances from the current literature
- which is the link among the hypotheses? give a short explanation
- can the Chinese lesson be exported abroad? Motivate
Round 2
Reviewer 1 Report
After carefully reading through the revised paper, I have one more comment: why the variable "resident" was removed from the model? Any reason behind this?
I am satisfied with the response to my comments.
Reviewer 2 Report
I accept all the changes made in the revised manuscript.
Still the description and significance of the three models explained in this paper is weak. Which model is the most significant and why?
